# The strength of balance: Strength and dynamic balance in children with and without hypermobility

Oluwakemi A. Ituen[1,2]*, Jacques Duysens[3], Gillian Ferguson[1], Bouwien Smits-Engelsman[1]

1 University of Uyo Teaching Hospital, Uyo, Akwa Ibom State, Nigeria, 2 Department of Health & Rehabilitation, University of Cape Town, Cape Town, South Africa, 3 Motor Control Laboratory, Movement Control and Neuroplasticity Research Group KU, Leuven, Belgium

* kemiituen@gmail.com

**Data Availability Statement:** All relevant data are within the manuscript and its Supporting Information files.

## Abstract

### Background

Generalized Joint hypermobility (GJH) is predominantly non-symptomatic. In fact, individuals with joint flexibility usually perform better than their non-hypermobile counterparts during physical activities. Notwithstanding, strength and balance are essential to maintain the control of the extra range of motion during activities and to prevent musculoskeletal complications. There are limited and conflicting pieces of evidence in literature regarding the association between strength and balance in children with GJH.

### Objectives

The purpose of this study was to examine differences in functional strength, dynamic balance, proprioception, and isometric strength in children with and without joint hypermobility and determine the association between strength outcomes and dynamic balance.

### Method

A cross-sectional study was conducted among children aged 6 to 11. Hypermobility was determined using the Beighton Score, with scores ≥6 representing hypermobility. Functional strength was assessed with the Functional Strength Measure (FSM), isometric strength was determined with a handheld dynamometer (HHD), the Y-Balance Test (YBT) was used to assess dynamic balance and the Wedges test to measure proprioception.

### Results

This study included 588 participants (age: 7.97 ± 1.3 years; height: 128±10.1 cm; mass: 27.18 ± 7.98 kg). 402 children were classified as having normal mobility and 186 as being hypermobile. Hypermobile children had better functional strength in the lower extremities than children with normal range mobility but lower reach distance in the YBT. No differences in proprioception, functional strength of the upper extremity or isometric strength in the hands were found. However, isometric lower extremity force was less in hypermobile

**Funding:** Oluwakemi A. Ituen (O.A.I.) research received grant from the Faculty of Health Sciences Faculty Research Committee of the University of Cape Town, South Africa, funding number 436359.

**Competing interests:** The authors have declared that no competing interests exist

**Abbreviations:** NM, Normal Mobility; HM, Hyper mobility; BMI, Body mass index; MANCOVA, Multivariate Analysis of Covariance; GJH, Generalized Joint Hypermobility; HSD, Hypermobility Spectrum Disorder; PARQ, Physical Activity Readiness Questionnaire; FSM, Functional Strength Measurement; WHO, World Health Organization; HHD, Hand-Held Dynamometer; FPS, Faces Pain Scale; Est, Estimated; RT, Right; LT, Left.

children than children with normal range mobility. Irrespective of their joint mobility, a fair significant correlation existed between total Y-balance distance and FSM items r = 0.16–0.37, p = 0.01. Correlations between total Y-balance distance and isometric strength of knee and ankle muscles ranged between r = 0.26–0.42, p = 0.001.

## Conclusion

Hypermobile joints seem to co-occur with lower extremity isometric strength, more functional strength in the lower extremities and less reaching distance in dynamic balance. The opposing direction of the results on functional and isometric strength tests highlights the importance of the type of outcome measures used to describe the association of strength and the range of motion.

## Introduction

Generalized Joint hypermobility (GJH) a result of laxity of ligaments, is commonly examined with the Beighton score, and its prevalence usually depends on age, gender and ethnicity [1,2]. GJH is typically of genetic origin but may also be acquired through exercises, stretching or trauma [3,4]. Although GJH enhances activities that require flexibility, it also poses risk for complications, specifically musculoskeletal symptoms [5–7]. The initial assumption by previous authors has been that a hypermobile joint is unstable, predisposing it to repetitive microtraumas that destroy mechanoreceptors over time [8,9]. This will lead to joint injury, arthralgia and other complications, such as compromised proprioception, impaired strength, and poor balance [10,11]. When GJH becomes associated with the aforementioned musculoskeletal symptoms it is referred to as Hypermobility Spectrum disorder (HSD) [12]. Even though GJH is a risk for developing musculoskeletal symptoms, biomarkers and clinical predictors of musculoskeletal symptoms are highly variable [13–15]. It is interesting that hypermobility is inherently more prevalent in children who are biologically immature when the growth of the musculoskeletal system is ongoing [13,16]. It remains a question to be answered if children with GJH will be more prone to micro trauma, because they are less coordinated or have less muscle power to adapt to sudden balance disturbances [17]. This raises the suggestion that immature muscle strength plays a role in GJH.

Strength and balance are important in the context of pathology [18]. They are essential for many daily and leisure activities, and it is assumed that a deficit of either will have a negative impact on an individual's participation levels [19]. Muscular fitness is a synergy of the different components of muscle activities (muscle strength, power and endurance) whereby multiple muscle groups work together in a coordinated way across a range of joint angles and, depending on the activity, for different periods [20–22]. Muscular strength is the maximum amount of force one can produce or the amount of weight one can lift [23] whereas, explosive power is the ability to generate a maximum muscular contraction instantly in a burst of movement [24]. On the other hand, the ability to repeat a movement for an extended period without fatiguing is muscle endurance [25,26]. Isometric strength is tested by a muscle contraction against maximum resistance over one joint in one direction with the rest of the body in a stabilized position [27]. Lastly, the strength needed to perform fundamental motor skills is called functional strength [26]. Yet, muscle strength in individuals with hypermobility has been mostly evaluated under isometric conditions, While functional strength may be more relevant

for daily activities [28,29]. Hence, the need to reevaluate the relationship between strength and GJH, as a greater functional strength compensates for the ligament laxity [30].

Balance is defined as the ability to maintain an upright posture and to keep the center of gravity within the limits of the base of support [31]. Muscle strength and proprioception have been reported to play significant roles in balance [32]. The most frequently used test to assess balance clinically in children with GJH is one leg stance [5,33]. Although, this test is sensitive in assessing balance, it is a static test and it also lacks task difficulty thus creating a ceiling effect [31,34]. In addition, testing balance dynamically is more appropriate because the need to keep your center of gravity within your limits of stability in order to prevent injuries is higher during functional activities than in quiet standing [35,36]. The Y-balance test, (standing on one leg and reaching forward as far as possible with the other leg) is a good alternative to the static one leg stance test as it induces large shifts in center of pressure unlike the small anticipatory shifts seen in static balance control [19]. Most importantly, the stability boundaries encountered during the reaching movement with the foot are very different from stationary upright one-leg standing making this task more challenging and less sensitive to a ceiling effect [35].

There are two possible routes for the onset of musculoskeletal symptoms in children with GJH [32]. First, stabilizing a joint during physical activities requires strength but when strength is compromised in a hypermobile joint, the possibility of sustaining injury is increased [37]. In another view, laxity of ligament or capsule may degrade the proprioceptive information from a hypermobile joint which may lead to delayed stabilization of the loaded joint inducing further damage of mechanoreceptors at the joint and this results in pain [38].

To disentangle this problem, the first step is to evaluate children with hypermobile joints before they have developed limiting musculoskeletal complaints and study if their proprioception, strength and balance are different from children with normal mobile joints. Next, one needs to examine the relationship between these outcomes and performance in a loaded dynamic balance task, imitating natural conditions. In our planned research, we will follow up on these children to see how many develop musculoskeletal complaints and which variable(s) predict the later development of these complaints.

In this study, we will answer the following research questions:

1. Are functional strength, dynamic balance, proprioception, or isometric strength diminished in a random sample of children with joint hypermobility between 6–11 years of age compared to children with normal range of joint motion?

2. How strong is the relationship between strength and balance in children between 6–11 years of age?

## Materials and method

### Subjects

The study used a cross-sectional descriptive design. The study was conducted following the Declaration of Helsinki. Ethical approval was obtained both from the human research ethics committee of the University of Cape Town (UCT HREC: 096/2015, HREC REF: 306/2021) and the University of Uyo Teaching Hospital REF: UUTH/AD/S/96/VOL/ XXI/524.The secretary of the Local Government education-Uyo, and the Anambra State Universal Basic Education Board Chairman, along with the head teachers and class teachers at the selected schools all granted permission to assess the children. Schools were selected through the convenience sampling method. The recruitment period was from 20th September 2021- 31st October 2021. The following exclusion criteria were applied: i) Children who have high risk level and poor

safety as it pertains to physical activity, this was assessed using The Physical Activity Readiness Questionnaire (PAR-Q) for children ii) Children who were limited in their ability to understand the testing instructions or the performance of the activities (e.g., cognitive impairment, gross motor impairment etc.) as reported by parents [39]. None of the children were excluded using the above criteria. The study sample size was calculated through a power analysis that showed that a total sample size of 164 per group was needed for a medium effect size (d = 0.4), at a power of 95%, while alpha is set at 0.05 with and allocation ratio of 1. The G-power analysis software version 3.1.9.2 was used for the sample size calculation [40]. Written informed consent was obtained from the parents or legal guardians of the children, and assent was given by the children before their enrolment. The children were tested by trained researchers in their school. The children were given breaks between tests or as requested by the child.

588 children were recruited, 186 were hypermobile. Children who reported with febrile illness on day of assessment were tested after recovery. Some of our study participants were not tested on Y-Balance (equipment was not available at the site at time of testing) and HHD (faulty equipment at the time of assessment). Halfway during the testing period, we observed the need to improve the sensitivity of the Wedges test and thus included additional wedges to achieve this (see flow chart in Fig 1). Only data with the more sensitive wedge test were included.

### Anthropometric measures

Data were collected on participants' age (years), sex, height (centimeters), and weight (kilograms). Height and weight were measured using measuring tape and weighing scale (on bare

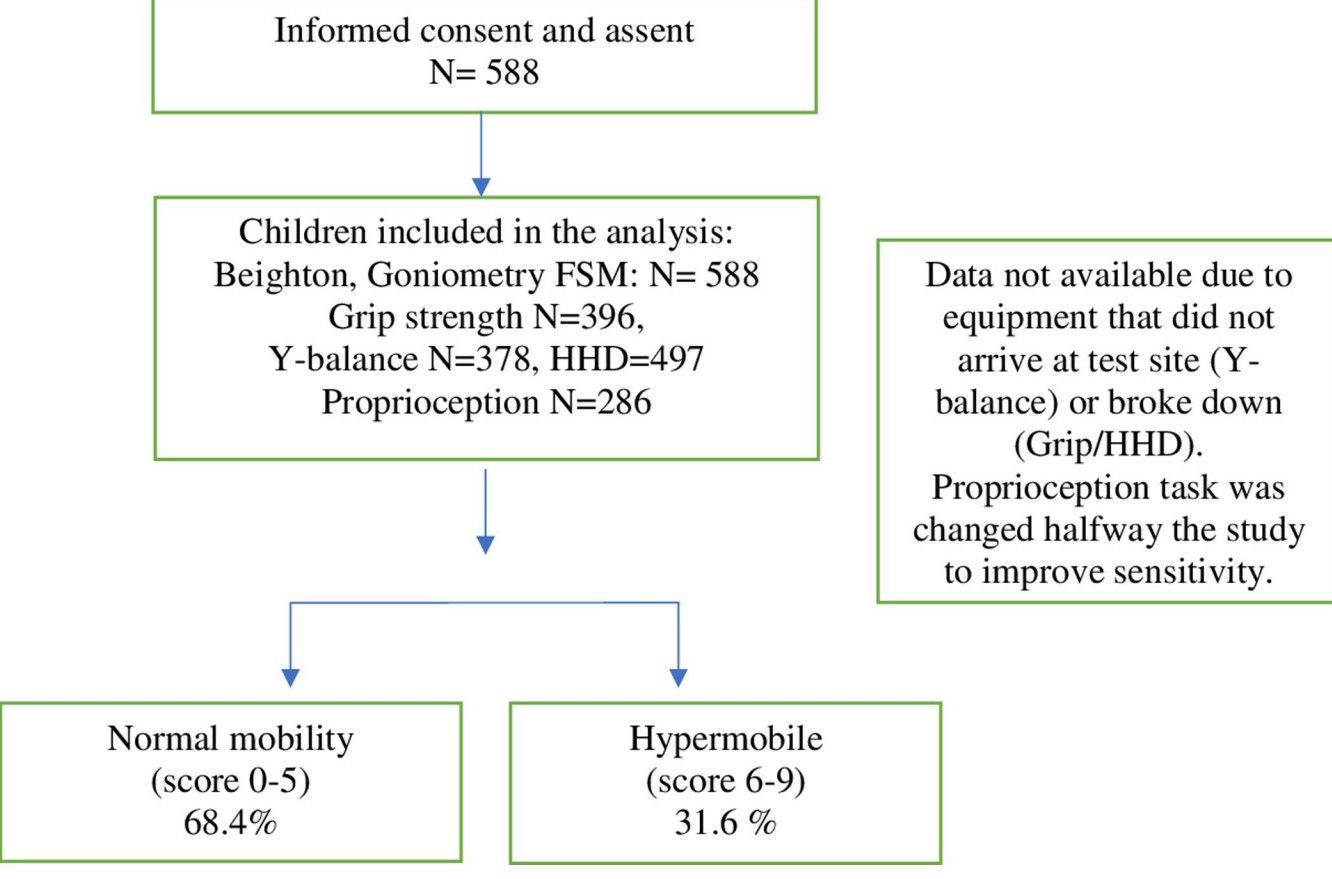

**Fig 1. Flow chart of study participants' recruitment.**

feet; measured to the closest one cm and 100 g, respectively). The body mass index (BMI) calculation was performed using a metric formula, weight (in kilograms) divided by height (in meters squared). Children were classified in Under-, Normal- and Overweight/Obese based on WHO norms for-age-and- sex [41].

## Beighton score

The nine-point Beighton score, with goniometry, was used to assess joint mobility [42]. The Beighton scoring system consists of bilateral assessment of the 5th metacarpophalangeal (MCP), elbow, knee joints, thumb movement and one active forward flexion task (Table 1). A score of 0–9 was used to divide joint mobility into two categories, normal mobility (0–5) and hypermobility (6–9) [43]. We established GJH with a Beighton score of $\geq 6$. The test has been validated among children [44].

## Pain questionnaire

The pain intensity was rated using Wong-Baker Faces Pain Scale (FPS) [45]. The FPS is a self-report pain scale that uses facial expressions to assess the intensity of pain. It is a valid instrument and has been frequently used to evaluate pain intensity in children [46,47].

## Functional Strength Measure (FSM)

The FSM is a comprehensive norm-referenced test for assessing functional strength in an activity [26]. The FSM test items comprise of two sets of four items in each set (upper limbs and lower limbs with four items in each set) [20]. There are the muscle power tests (e.g., standing long jump, overarm throwing) and muscle endurance tests (e.g., sit to stand, lifting a box). For items description see Table 2. The FSM has been validated among different groups of children and satisfactory values of test retest reliability of 0.91–0.94 were found [26].

## Hand held dynamometer

The Hand-Held Dynamometer (HHD) was used to assess maximum isometric muscle contraction of the knee extensors, ankle extensors and flexors and grip strength [48]. To measure the isometric strength of the knee extensors, the participants were seated in a table with the knees at 90˚ of flexion and the HHD placed on the anterior surface of the leg. The plantar flexors and dorsiflexors, were assessed in supine position, the HHD was placed on the sole of the foot to test for plantar flexors and the dorsum of the foot to test for dorsiflexors. The break method was used and the best of three trials was used for the analysis [20]. The HHD is a reliable instrument that has been used in children with ICC values ranging from 0.73 to 0.99 [49].

**Table 1. Beighton scoring system.**

| Items | Right | Left |
|---|---|---|
| a. Passive opposition of the thumb to the volar side of the forearm | 1 | 1 |
| b. Passive dorsiflexion of the 5th MCP joint to $\geq 90˚$ | 1 | 1 |
| c. Passive hyperextension of the elbow joint to $\geq 10˚$ | 1 | 1 |
| d. Passive hyperextension of the knee joint to $\geq 10˚$ | 1 | 1 |
| e. Placing hands flat on the floor with the knees kept straight | 1 | |
| Maximum possible score | 9 | |

**Table 2. Items of the functional strength measurement.**

| Items of the FSM | Item description |
|---|---|
| **Overarm throwing (cm)** | Throwing a heavy bag as far as possible. |
| **Standing long jump (cm)** | Jumping forwards as far as possible. |
| **Underarm throwing (cm)** | Throwing a heavy bag as far as possible. |
| **Chest pass (cm)** | Pushing a heavy bag as far as possible. |
| **Lateral step up (# in 30 s)** Tested on right and left | Touch the foot to the floor as fast as possible while standing on one leg on the lowest step of the stairs. |
| **Sit to stand (# in 30 s)** | Stand up and sit down as quickly as possible. |
| **Lifting a box (# in 30 s)** | Lift a plastic box filled with heavy bags onto a wooden box |
| **Stair climbing (# in 30 s)** | Climbing up and down stairs as quickly as possible |

## Y-Balance

The Y-balance test was developed as a time efficient test to replace the Star Excursion Balance Test (SEBT) [50]. It assesses dynamic stability in three directions (anterior, posteromedial, and posterolateral) instead of eight as was the case in SEBT. The test kit consists of a platform for stance and three pipes connected to the stance platform. The subject to be tested stands on the stance platform and uses the other limb to move the reach indicator along the calibrated pipe. The distance reached (measured in centimeters) is recorded. The Y-Balance test is considered effective in predicting injuries [51]. The Y-Balance test has demonstrated an excellent interrater reliability within session (ICC > 0.995) and between sessions (0.907 ≤ ICC ≤ 0.974) among children [19]. The test is performed three times and the best of the three is normalized with body height or limb length because of their reported correlation [52]. Limb length was not collected at all schools, while height was, hence we tested the relation between normalized distance using limb length and by using height for the 195 children for which we had both. Results showed correlations between normalized distances calculated based on limb length or height to be 0.93 both for right and left leg. This confirmed that it was valid to use normalization with height for all children in our analysis.

## Wedges test

We tested proprioception (detection of heel-height difference) using the wedges of various heights that produce different angles equal in surface, 1.5˚, 3˚, .4.5˚, 6˚, 9˚ and 12˚. The 1.5˚, 4.5˚ wedges were added to have more combinations with only 1.5˚ difference (1.5˚ versus 3˚, 3˚ versus 4.5˚, 4.5˚ versus 6˚).

Wedges demonstrated ecological validity in a study that measured proprioception among 7–10 years old Nigerian children with GJH [53]. Tools that have ecological validity can show the role proprioception plays in physical activities [54].

Participants stood behind a table and were not blindfolded during the testing but were instructed to not look at their feet while the test was conducted. While standing on the wedges, (without support from the table) they raised the arm of the side with the higher ankle. For example, the right arm for the right ankle. Both arms were raised when no difference in ankle-height was detected. The subject had 5 seconds to respond. A penalty score was awarded to every incorrect response, and it was determined by differences in the height of the wedges. The

higher the wedges height difference, the higher the penalty score. The individual penalty scores were summed up to get a total penalty; a high penalty score indicates poor proprioception

## Data analysis

Data was analyzed using the SPSS version 28. Descriptive statistics (frequency, percentage, mean, and standard deviation) were used to present the demographic data, Beighton scores, and Beighton classification of the study population. FSM, HHD, Y-balance and Wedges data were checked for outliers using histograms and z-score. Data points with z-values of more than 3.29 were deleted. No more than 3 data points per variable were removed. Because of the large differences on demographic variables between the normal and hypermobile children, all tests were adjusted for age, weight, and height. As a next step, Box test of equality of covariance matrixes was examined and found significant; thus, Pillai's Trace will be reported. No violation of linearity between the dependent variables and covariates was found. Seven multivariate analyses of covariance (MANCOVA) were conducted with Beighton classification group as independent variable, and functional strength of lower extremity, upper extremity, Y-balance, isometric strength knee, grip force, ankle strength and proprioception were treated as sets of dependent variables, concerning the same construct and with the same number of data entries. Two-tailed partial correlation analyses, controlling for age, were performed to verify the relationship between strength and Y-balance. Alpha was set at 0.01.

## Results

### Participant demographic and anthropometric characteristics

Our study included 588 children, 281 males and 307 females. The study mean age was 7.97 (S. D 1.3) years. Most children were normal weight (61.6%), 22.8% were underweight and 15.6% were overweight/obese.

### Joint mobility

Of the total sample, 402 children were classified as normal mobile (male, female) and 186 children (79 male and 107 female) were classified as hypermobile. Although the two groups came from a one convenience sample (same schools, same background) the two classification groups showed some differences. The children with hypermobility were significantly younger, smaller, and lighter (p = 0.001) than the children with normal mobility (see details in Table 3). This was corrected for in the statistics.

**Table 3. Mean (standard deviation) of the demographic and anthropometric data for the two groups of participants.**

| Demography<br>N | Normal mobile<br>402 | Hypermobile<br>186 | p-value |
|---|---|---|---|
| Age (Years) | 8.2 (1.3) | 7.5 (1.2) | 0.001 |
| Height (cm) | 1.29 (0.1) | 1.24(0.1) | 0.001 |
| Limb length (cm) | 129.9 (10.2) | 124.4 (8.8) | 0.001 |
| Weight (kg) | 28.6 (8.34) | 24.1 (6.0) | 0.001 |
| BMI (kg/m$^2$) | 16.7 (3.2) | 15.4 (2.8) | 0.001 |

N = Number of participants.

## Joint hypermobility and pain

In this random sample, 498 children reported no pain, and 90 children (71 normal mobile, 19 hypermobile) reported some pain. Of these 90, more children belonged to the normal mobile children (n = 71, 78.9%) than to the group children with hypermobility (n = 19, 21.1%) ($X^2$ = 3.15, p 0.08). However, this difference was not statistically significant. None of the children reported long term pain or pain during the testing.

## Group differences in Strength: Functional (FSM), and isometric strength (HHD)

The multivariate analysis showed that hypermobile children had better outcomes on the FSM lower extremity scores (p = 0.001). The univariate analysis showed significant differences for all items except for stairs (p = 0.19). For the multivariate outcomes of the upper limb items of the FSM, no significant differences were found. Estimated means were corrected for age, weight, and height and the statistics are presented in Table 4.

The isometric strength of the knee extensors, and ankle flexors and extensors, and grip force were measured using the HHD. The children with normal mobility had significantly higher mean values for the knee extensors (Fig 2), ankle extensors and ankle flexors but not for grip strength. Details of isometric strength outcomes for the two groups and statistics are presented in Table 5.

## Group differences in balance and proprioception

The multivariate analysis revealed that study participants with normal mobility reached further on the Y-Balance test. Univariate analysis showed that this was the case for posteromedial and posterolateral directions, but no differences were found in anterior direction.

No differences were found between groups on the proprioception outcomes. A high penalty or low correct score on the wedges' tests indicated poor proprioception (Estimated Means and statistics are shown in Table 5).

## Association between strength and balance

Partial correlation, controlled for age, was performed and moderate partial correlations were found between the Y-balance mean score and FSM strength throw outcomes of the Upper extremity and isometric strength of knee and ankle muscles (Table 6). No significant correlations were found between the lower extremity items: long jump and sit to stand and Y-balance total score.

# Discussion

Adequate muscle strength, power, and endurance enhances participation in daily activities and sports, especially in children with hypermobile joints as they will require more strength to control the extra range of motion. In addition, good joint stability and balance are important in the prevention of injuries during physical activities [8]. Therefore, the aim of this study was to examine the relationship between joint mobility and different aspects of strength, proprioception, and dynamic balance.

The prevalence of GJH has been reported to be higher among children, females and Africans [55] but there is a lack of consensus on the Beighton score cut off [56,57]. This accounts for the variation seen in the reported prevalence of GJH. According to some authors a Beighton score cut off of ≥4 will result in an overrepresentation of GHJ among children who are predominantly hypermobile [55,58]. This justifies our use of Beighton score cut off of ≥6

**Table 4. Estimated means (corrected for age, height, and weight) for the two joint mobility groups, on functional strength and isometric strength.**

| | Normal mobile Est. Mean (Std. error) | Hypermobile Est. Mean (Std. error) | F-value | P-value | Partial eta squared |
|---|---|---|---|---|---|
| **FSM n = 588** | | | | | |
| **MANCOVA Lower limbs** | | | | | |
| Beighton classification[a,b,c] | | | 7.7 | 0.001 | 0.062 |
| **Univariate** | | | | | |
| **Lower limbs** | | | | | |
| Lat. Step up RT | 37.6 (0.7) | 43.6 (1.0) | 22.4 | 0.001 | 0.037 |
| Lat. Step up LT | 37.8 (0.7) | 43.3 (1.0) | 19.6 | 0.001 | 0.033 |
| Stair climbing | 66.5 (0.5) | 65.2 (0.8) | 1.7 | 0.19 | 0.003 |
| Sit to stand | 24.0 (0.3) | 26.7 (0.8) | 18.4 | 0.001 | 0.031 |
| Long jump | 111.8 (1.3) | 118.6 (1.9) | 7.9 | 0.005 | 0.013 |
| | | | | | |
| **FSM n = 496** | | | | | |
| **MANCOVA Upper limbs** | | | | | |
| Beighton classification[a,b,c] | | | 3.6 | 0.001 | 0.063 |
| **Univariate** | | | | | |
| **Upper limbs** | | | | | |
| Upper hand throw | 200.2 (2.7) | 199.8 (3.7) | 0.01 | 0.92 | 0.0001 |
| Under hand throw | 258.9 (4.2) | 251.0 (5.8) | 1.2 | 0.28 | 0.002 |
| Chest pass | 160.6 (2.2) | 160.7 (3.0) | 0.0001 | 0.99 | 0.0001 |
| Lifting of box | 16.8 (0.3) | 16.4 (0.4) | 0.46 | 0.50 | 0.001 |
| **HHD n = 487** | | | | | |
| **MANCOVA Knee extensors** | | | | | |
| Beighton classification[b,c] | | | 11.7 | 0.001 | 0.046 |
| **Univariate** | | | | | |
| Knee extensors RT | 122.8 (1.7) | 107.1 (2.7) | 23.4 | 0.001 | 0.046 |
| Knee extensors LT | 117.2 (1.6) | 104.5 (2.6) | 17.1 | 0.001 | 0.034 |
| | | | | | |
| **HHD n = 396** | | | | | |
| **MANCOVA Grip strength** | | | | | |
| Beighton classification[a] | | | 0.26 | 0.773 | 0.001 |
| **Univariate** | | | | | |
| Grip RT | 46.5 (0.8) | 45.5 (1.1) | 0.503 | 0.479 | 0.001 |
| Grip LT | 43.5 (0.7) | 42.6 (1.1) | 0.384 | 0.536 | 0.001 |
| | | | | | |
| **HHD n = 287** | | | | | |
| **MANCOVA Ankle muscles** | | | | | |
| Beighton classification[b,c] | | | 2.9 | 0.01 | 0.058 |
| **Univariate** | | | | | |
| Dorsiflexors RT | 83.2 (2.0) | 74.0 (2.6) | 7.3 | 0.007 | 0.025 |
| Dorsiflexors LT | 85.3 (2.8) | 71.3 (3.7) | 8.8 | 0.003 | 0.033 |
| Plantarflexors RT | 97.2 (2.2) | 87.4 (2.9) | 7.1 | 0.008 | 0.025 |
| Plantarflexors LT | 98.0 (2.0) | 87.5 (2.6) | 9.5 | 0.002 | 0.030 |

Significant covariates: a = height, b = weight, c = age. RT = Right, LT = Left, n = number of participants.

in this present study, although it can be considered strict and age specific when compared to previous studies that used Beighton score cut off of $\geq 4$ to identify GJH among children [4,59]. It is interesting that even with our strict Beighton score cut off point, we found a high prevalence of GJH (31%) in our sample compared to outcomes from previous studies among Caucasians [28,60]. This further justifies the plea for the higher Beighton score cut off among children [61]. Although this has not been confirmed in other studies, Sohrbeck-Nøhr 2014 found children with Beighton score of 5 or 6 having greater odds of developing musculoskeletal complaints than those with Beighton score of 4 [62]. It appears that the number of hypermobile joints a child has may be a factor in the onset of musculoskeletal complaints hence the need for higher Beighton score cut off in identifying GJH.

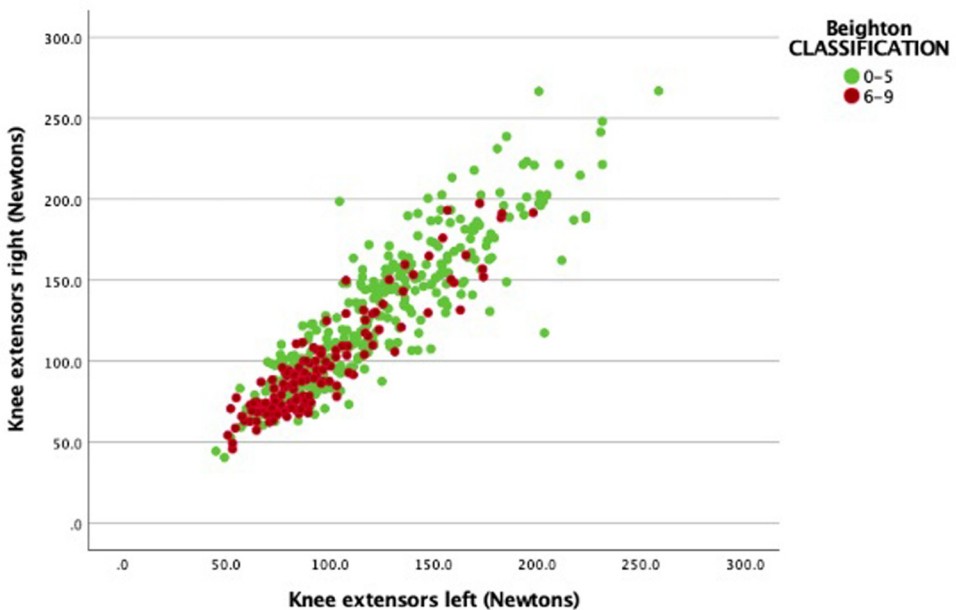

**Fig 2. Means isometric knee extensor strength values and Beighton classification.**

The children in the hypermobile group were significantly younger and with more females which is in line with previous studies [43,55]. Nevertheless, these differences could not explain the results and group differences persisted after correction. Even though GJH is not a disease condition, musculoskeletal complications do arise when individuals with GJH cannot control the extra degrees of motion during physical activities [63]. The effects of these musculoskeletal complications and cost of rehabilitation are far reaching [46]. This implies that understanding the pathway to the onset of musculoskeletal complications in GJH and tailoring it towards prevention and prompt management cannot be overemphasized.

**Table 5. Estimated means (corrected for age, height, and weight) for the two joint mobility groups, on balance and proprioception.**

|  | Normal mobile Est. Mean (Std. error) | Hypermobile Est. Mean (Std. error) | F-value | P-value | Partial eta squared |
|---|---|---|---|---|---|
| **Y-Balance test (n = 378)** | | | | | |
| **MANCOVA Balance** | | | | | |
| Beighton classification[a,b] | | | 4.6 | *0.001* | 0.070 |
| **Univariate** | | | | | |
| Anterior RT | 39.6 (0.5) | 40.1 (0.6) | 0.3 | 0.6 | 0.001 |
| Anterior LT | 40.3 (0.5) | 40.9 (0.6) | 0.6 | 0.4 | 0.002 |
| Posterior-medial RT | 46.9 (0.6) | 44.1 (0.8) | 7.6 | 0.006 | 0.020 |
| Posterior-medial LT | 48.6 (0.7) | 45.1 (0.8) | 11.1 | 0.001 | 0.029 |
| Posterior-lateral RT | 47.9 (0.6) | 44.9 (0.8) | 8.5 | 0.004 | 0.023 |
| Posterior-lateral LT | 49.2 (0.7) | 45.8 (0.8) | 9.7 | 0.002 | 0.026 |
|  | | | | | |
| **Proprioception n = 286** | | | | | |
| **MANCOVA Wedges** | | | | | |
| Beighton classification | | | 0.9 | *0.42* | 0.006 |
| **Univariate** | | | | | |
| Total penalty score# | 4.1 (0.4) | 3.4 (0.4) | 1.8 | 0.183 | 0.006 |
| Total correct score | 18.6 (0.2) | 18.8 (0.2) | 1.0 | 0.328 | 0.004 |

Significant covariates: a = height, b = weight. RT = Right, LT = Left, n = number of participants.

**Table 6. Partial correlation between Y-Balance and functional strength and isometric muscle strength.**

|  | Total Beighton r (p) | Mean Normalized Y-Balance r (p) | Normalized Y-Balance Right r (p) | Normalized Y-balance Left r (p) |
|---|---|---|---|---|
| FSM n = 588 Lower limbs |  |  |  |  |
| Lat. Step up RT | 0.223 *** | 0.157 ** | 0.167 *** | 0.141** |
| Lat. Step up LT | 0.223 *** | 0.160 ** | 0.168 *** | 0.144** |
| Stair climbing | -0.150 *** | 0.256 *** | 0.238 *** | 0.263*** |
| Sit to stand | 0.238 *** | ns | ns | ns |
| Long jump | 0.214 *** | ns | ns | ns |
| FSM n = 496 Upper limbs |  |  |  |  |
| Upper hand throw | ns | 0.359 *** | 0.341*** | 0.362*** |
| Under hand throw | ns | 0.366 *** | 0.345*** | 0.371*** |
| Chest pass | ns | 0.165 ** | 0.168** | 0.156** |
| Lifting of box | ns | 0.175 ** | 0.18 ** | 0.158** |
| HHD n = 487 Knee extensors |  |  |  |  |
| Knee extensors RT | -0.418 *** | 0.342 *** | 0.300 *** | 0.364 *** |
| Knee extensors LT | -0.372 *** | 0.259 *** | 0.226 *** | 0.275 *** |
| HHD n = 396 Grip strength |  |  |  |  |
| Grip RT | -0.1* | ns | ns | ns |
| Grip LT | ns | ns | ns | ns |
| HHD n = 287 |  |  |  |  |
| Dorsiflexors RT | -0.402 *** | 0.402 *** | 0.373 *** | 0.408 *** |
| Dorsiflexors LT | -0.434 *** | 0.418 *** | 0.382 *** | 0.429 *** |
| Plantar flexors RT | -0.344 *** | 0.303 *** | 0.274 *** | 0.313 *** |
| Plantar flexors LT | -0.330 *** | 0.335 *** | 0.312 *** | 0.338 *** |

***Correlation is significant at the 0.001 level (2-tailed)

** Correlation is significant at the 0.01 level (2-tailed), *Correlation is significant at the 0.05 level (2-tailed), ns = not significant, RT = Right, LT = Left, n = number of participants.

## Pain and GJH

In this present study, pain was reported by about 18% of the children but this was unrelated to GJH. In addition, none of the children with GJH had reported pain in three joints or over three months duration nor during performance of the test items. Even though we probed children to discuss their joint pain while completing the questionnaire, it did not seem to hamper them in the activities. Pain can be a feature of musculoskeletal disorder, but it is not commonly reported in children with GJH [64]. Previous studies have not been able to establish an association between joint hypermobility and pain because children may report pain irrespective of their joint mobility status [2,65]. It is therefore imperative to look beyond pain in an attempt to find out why some children with GJH are liable to severe musculoskeletal complaints and injuries later in life [2]. Although altered proprioception, lower muscle strength, joint instability, dislocation, and other musculoskeletal impairments have been reported in children and adults with HSD, this trend has not been confirmed in children with GJH [66].

## Proprioception and GJH

Proprioception was examined using the wedges test. Although no significant differences were found, children with GJH did not perform worse than children with normal mobility. If

anything, they made less mistakes (lower penalty score). This outcome is in line with the study by Ituen et al, and further confirms that joint laxity may not compromise proprioceptive acuity in young children with GJH [53]. Evidence in literature has supported the notion of joint pain because of the relation between poor motor performance and the occurrence of injury [67]. Since joint pain was unrelated with GJH in our study, it was to be expected that the children with GJH had a good functional status.

### Functional strength and GJH

In this study, the children with GJH had a significantly better performance in the lower limb items of the FSM except for stair climbing, whereas no differences were found in the upper limb items of the FSM. This difference is important as injuries are expected to happen more often in the loaded positions, thus in lower limbs [68]. An association of leg musculoskeletal symptoms and GJH, has been inconclusive in the literature because the musculoskeletal symptoms among individuals with GJH are variable and temporary [67,69]. Furthermore, bony structures in the lower limbs provide stability that may reduce the negative effect of hypermobility [28].

The outcome of lower limb functional strength in children with GJH in our study is comparable to the study by Juul-Kristensen et al [70]. In their study, children with GJH had better peak vertical jump displacement than children with normal mobility. Junge et al, also tested jump distance both in children with GJH and in those with normal mobility and they did not find a difference in their performances [71].

The efficiency of leg movements is based on contributions from both the passive component (ligaments) and active component (muscles) [72]. So, a closer look at muscle activation in individuals with GJH will help us to understand how they are able to move effectively despite the extra degrees of movement at the joint. It is known that muscle activation compensatory strategies can overcome consequences of joint dysfunction [71]. This compensation occurs either by activation of other muscle groups or by co-contraction of muscles, providing the joint stabilization necessary during functional movement and thus preventing injuries [71]. It has been demonstrated in previous studies that neuromuscular strategies in individuals with GJH are different from those with normal mobility [73,74]. For instance, the ankle is considered to be overactive in children with GJH as studies on electromyography have found activation of ankle muscles to be significantly higher in children with GJH [7], whereas the ankle strategy is only utilized in children with normal mobility during very difficult tasks. In performing the single leg hop test, Junge et al found that children with GJH have an increased activation of the gastrocnemius muscles, which was significantly different from children with normal mobility [71]. Increased co-contraction of muscles is another strategy used by children with GJH to stabilize the joint for better functional outcomes [73]. This was also evident in the study by Greenwood et al, although their study population consisted of children with Benign Joint Hypermobility Syndrome (BJHS) [74]. The co-contraction of rectus femoris and semitendinosus was higher in children with BJHS than in the control group. How effective these strategies are in the long run, given the lower isometric strength level and the increased risk of fatigue is still unknown.

The only lower limb FSM item in which children with GJH did not outclass the children with normal mobility was stairclimbing. Some authors are of the opinion that unlike level walking, stair climbing requires quick stabilization of joints and fast activation of muscles, so it will be more difficult than level walking [72,75]. In fact, stair climbing may therefore provide a prodromal impairment in GJH [12] The knee joint is very significant in stair climbing, and quadriceps acts as joint stabilizer during that activity. It should be emphasized that stair

climbing was still at a high level in GJH. Thus, the lack of significant reduction in scores on stair climbing reflects, that even with lower isometric extensor force, the children successfully compensated with muscle activation for the laxity of the joints. However, this may change when they grow older.

## Isometric strength and GJH

Significantly lower isometric strength (quadriceps, plantar flexors and dorsiflexors) was measured among children with GJH, however, grip strength was not different between groups. Contrary to our findings, Jensen et al., reported no difference in isometric knee strength between children (10 years) and the healthy control in their study [73]. Typically, the presence of pain precedes reduction in physical activities and consequently deconditioning and muscle weakness [76]. However, Scheper et al., argued that muscle weakness in asymptomatic individuals with GJH may not necessarily be only as a consequence of deconditioning secondary to pain and inactivity [69]. They were of the opinion that the elasticity of tendons also influences the amount of force a muscle can generate. In agreement with this, some authors have suggested that structural changes such as reduced tendon stiffness, similar to what is seen in people with connective tissue disorder, may be a contributing factor to muscle weakness in GJH [70,71]. In addition, the possibility of fear avoidance and anxiety because of excess movement and injury exists but this is yet to be explored in children with GJH. The association between GJH and anxiety has been confirmed in adolescents and young adults with GJH

## Dynamic balance and GJH

While keeping balance on one leg, children with GJH in our study could not reach as far as those with normal mobility in posterolateral and posteromedial directions, but they performed equally in the anterior direction. Trunk movement and stability of the knee joint is essential during this test. Children with GJH often try to hyperextend their knee to obtain stability. However, to reach further backwards when pushing against the indicator either in medial or lateral posterior direction, the knee of the stance leg needs to flex. Moreover, the trunk needs to lean forward to counterbalance for the back leg in order to keep balance and prevent a fall. Thus medio-lateral stability is challenged in these items in a flexed position of the knee, which might reveal the limits of their compensatory stability.

Even though their study was in an adult population, Hou et al., evaluated balance in individuals with chronic instability using the Star Excursion Balance Test [63]. The authors reported an initial deficit in posterolateral and posteromedial directions in the group with GJH which improved following a balance training program.

## The association between strength and balance

When we consider that joint stability is achieved passively by ligaments and actively by muscle contraction, then the association between strength and balance in GJH is very important. Based on our hypothesis, we expected a strong association between strength and balance, yet our study found a weak (FSM upper extremity) or moderate (FSM lower extremity) correlation with isometric strength when we controlled for age. In contrast, the study by Hou et al. demonstrated a close relationship between balance and strength in hypermobile adults with chronic ankle instability [63]. They found an improvement in balance and a corresponding gain in muscle strength following balance training. The discrepancy between Hou et al. and this study needs exploring. An advantage in the present sample, compared to individuals with chronic ankle instability, is the good level of proprioception. Proprioception is very important in motor control as it facilitates prompt activation of muscle. It is likely that the better lower

limb performance on FSM that we found among the children with GJH may be the result of their proprioceptive acuity and lifelong exposure to their joint laxity during daily activities

An unexpected finding was the relative high correlations between reaching distance in the Y-balance and the explosive power items of the FSM.

At first sight upper body strength seems less important to keep balance in one leg stance. However, in the Y-balance tasks, reaching the limits of stability depends on moving the upper body as far and steady forwards and backwards to counterbalance the leg movement in the other direction. This might explain the observed association (r = 0.36), which is in the same range as the relation with knee (r = 0.34, r = 0.26) and ankle muscle strength (r = 0.30-r = 0.42) for right and left leg respectively.

## Strengths and limitations of study

In this study, a strict Beighton cut-off of $\geq 6$ as recommended in literature was used, which makes our study population with GJH highly mobile. Even though we had a large sample size, some of the children did not take part in all tests. However, we are confident that the remaining number of participants was high enough to support the claims made.

Our study has also provided data on functional tasks like stair climbing or lifting a heavy box in children with GJH, which is scarce in literature, making comparisons with other studies difficult. Clinically, tests of dynamic balance and functional strength should be included in assessment protocol because they are more related to activities of daily living. Also our study provides evidence of stair climbing as a prodromal impairments for children with GJH. In addition the items of the FSM are more cost effective and easily assessable than the HHD. However, we believe that the assessment of dynamic balance and functional strength is relevant to our activities of daily living. The measurement of isometric strength was done with HHD, which makes it objective, but the method has its limitations.

## Conclusion

Although functional strength was significantly higher in the lower limbs of children with GJH, this study showed impaired isometric strength in their lower extremity. Our results indicate that hypermobility did not compromise proprioception and functional motor performance because the children may have established compensatory strategies to cope with their extra range of motion. The value of isometric force may be overestimated, since the present results clearly demonstrate that force, measured as functional strength, is not decreased in this population. However, the long-term effect of these compensatory strategies on the musculoskeletal system still needs to be discovered.

## Supporting information

**S1 File. PLOSONE1.**
(ZIP)

## Acknowledgments

We acknowledge the support of parents, children, management, and teachers at the participating schools. Furthermore, we acknowledge Dr Christie Akwaowo, Department of community medicine, University of Uyo, Ijeoma Blessing Anieto, Ebuka Anieto (physiotherapists), and physiotherapy students from Nnamdi Azikiwe University, Nigeria for their help with the data collection.

## Author Contributions

**Conceptualization:** Oluwakemi A. Ituen, Jacques Duysens, Gillian Ferguson, Bouwien Smits-Engelsman.

**Funding acquisition:** Gillian Ferguson.

**Methodology:** Oluwakemi A. Ituen, Jacques Duysens, Bouwien Smits-Engelsman.

**Supervision:** Bouwien Smits-Engelsman.

**Writing – original draft:** Oluwakemi A. Ituen.

**Writing – review & editing:** Jacques Duysens, Gillian Ferguson, Bouwien Smits-Engelsman.

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
