## [Decision Letter · Decision Letter 0]

5 Feb 2024

PONE-D-24-00228The strength of balance: Strength and balance in children with and without hypermobilityPLOS ONE

Dear Dr. Ituen,

Thank you for submitting your manuscript to PLOS ONE. After careful consideration, we feel that it has merit but does not fully meet PLOS ONE’s publication criteria as it currently stands. Therefore, we invite you to submit a revised version of the manuscript that addresses the points raised during the review process.

We look forward to receiving your revised manuscript.

Kind regards,

Mehrnaz Kajbafvala, Ph.D

Academic Editor

PLOS ONE

Reviewers' comments:

Reviewer's Responses to Questions

**Comments to the Author**

1. Is the manuscript technically sound, and do the data support the conclusions?

Reviewer #1: Yes

Reviewer #2: Yes

2. Has the statistical analysis been performed appropriately and rigorously? 

Reviewer #1: Yes

Reviewer #2: Yes

3. Have the authors made all data underlying the findings in their manuscript fully available?

Reviewer #1: Yes

Reviewer #2: Yes

4. Is the manuscript presented in an intelligible fashion and written in standard English?

Reviewer #1: Yes

Reviewer #2: Yes

5. Review Comments to the Author

Reviewer #1: This study was aimed to examine differences in functional strength, dynamic balance, proprioception, and isometric strength in children with and without joint hypermobility and determine the association between strength outcomes and dynamic balance. The study findings indicated that Hypermobile joints seem to co-occur with lower extremity isometric strength, more functional strength in the lower extremities and less reaching distance in dynamic balance. The opposing direction of the results on functional and isometric strength tests highlights the importance of the type of outcome measures used to describe the association of strength and the range of motion. Overall, the study is interesting, however there are several clarifications needed.

Comment#1

Title. Considering the explanations provided in the Introduction section and variables assessed, the title is not appropriate. Please edit it.

Comment#2

Please add “pain” and “dynamic balance” to the Keywords section.

Comment#3

Introductions. Please insert reference/references for different sentences in the Introduction section. For some sentences, no reference was used.

Comment# 4

Introduction, please summarize the Introduction section.

Reviewer #2: Thank you for the opportunity to review this manuscript. The manuscript is well-written and detailed, I thank the research team for their efforts. Comments are provided below.

The introduction in its present form is too long. It distracts the reader from the main topic. Please summarize it.

Some of the sentences in the introduction have no references. Please insert them.

Table 6 is inserted in the manuscript text.

The clinical implication of this study is added.

6. PLOS authors have the option to publish the peer review history of their article (what does this mean?). If published, this will include your full peer review and any attached files.

Reviewer #1: No

Reviewer #2: No

---

## [Author Response · Author response to Decision Letter 0]

8 Mar 2024

Reviewer 1

1 Title. Considering the explanations provided in the Introduction section and variables assessed, the title is not appropriate. Please edit it.

 Thank you for raising this point. The title has been changed to reflect the context of the article. 

The strength of balance: strength and dynamic balance in children with and without hypermobility.

2 Please add “pain” and “dynamic balance” to the Keywords section.

 Thank you for pointing this out. Pain and dynamic balanced are added to the keywords.

Keywords: Generalized Joint Hypermobility, Strength, Dynamic balance, Children, Proprioception, and Pain.

3 Introductions. Please insert reference/references for different sentences in the Introduction section. For some sentences, no reference was used 

Thank you for raising this point. All sentences are now referenced excepts are those that are the opinions of the authors 

4 Introduction, please summarize the Introduction section. 

Thank you for this comment. The introduction has been summarized and the word count reduced from 1222 words to 889 words. Some sentences were adapted to ensure connection (flow) with the text after reducing the number of words

REVIEWER 2

S/N Original manuscript Revised manuscript

1 The introduction in its present form is too long. It distracts the reader from the main topic. Please summarize it.

Thank you for this comment. The introduction has been summarized and the word count reduced from 1222 words to 889 words. Some sentences were adapted to ensure connection (flow) with the text after reducing the number of words

2 Some of the sentences in the introduction have no references. Please insert them. 

 Thank you for raising this point. All sentences are now referenced except those that are the opinions of the authors.

3 Table 6 is inserted in the manuscript text. Thank you for pointing this out. The insertion of Table 6 has been adapted.

4 The clinical implication of this study is added.

 Thank you for this comment. This has been included in the strength and limitations section of the manuscript. “Our study has also provided data on functional tasks like stair climbing or lifting a heavy box in children with GJH, which is scarce in literature, making comparisons with other studies difficult. Clinically, tests of dynamic balance and functional strength should be included in assessment protocol because they are more relatable to activities of daily living. Also our study provides evidence of stair climbing as a prodromal impairments for children with GJH. In addition the items of the FSM are more cost effective and easily assessable than the HHD. The measurement of isometric strength was done with HHD, which makes it objective, but the method has its limitations. “

---

## [Decision Letter · Decision Letter 1]

1 Apr 2024

The strength of balance: Strength and dynamic balance in children with and without hypermobility

PONE-D-24-00228R1

Dear Dr. Oluwakemi Adebukola Ituen,

We’re pleased to inform you that your manuscript has been judged scientifically suitable for publication and will be formally accepted for publication once it meets all outstanding technical requirements.

Kind regards,

Mehrnaz Kajbafvala, Ph.D

Academic Editor

PLOS ONE

Additional Editor Comments (optional):

Reviewers' comments:

Reviewer's Responses to Questions

**Comments to the Author**

1. If the authors have adequately addressed your comments raised in a previous round of review and you feel that this manuscript is now acceptable for publication, you may indicate that here to bypass the “Comments to the Author” section, enter your conflict of interest statement in the “Confidential to Editor” section, and submit your "Accept" recommendation.

Reviewer #1: All comments have been addressed

Reviewer #2: (No Response)

2. Is the manuscript technically sound, and do the data support the conclusions?

Reviewer #1: Yes

Reviewer #2: (No Response)

3. Has the statistical analysis been performed appropriately and rigorously? 

Reviewer #1: Yes

Reviewer #2: (No Response)

4. Have the authors made all data underlying the findings in their manuscript fully available?

Reviewer #1: Yes

Reviewer #2: (No Response)

5. Is the manuscript presented in an intelligible fashion and written in standard English?

Reviewer #1: Yes

Reviewer #2: (No Response)

6. Review Comments to the Author

Reviewer #1: (No Response)

Reviewer #2: (No Response)

7. PLOS authors have the option to publish the peer review history of their article (what does this mean?). If published, this will include your full peer review and any attached files.

Reviewer #1: No

Reviewer #2: No

---

## [Editor Report · Acceptance letter]

26 Apr 2024

PONE-D-24-00228R1 

PLOS ONE

Dear Dr. Ituen, 

I'm pleased to inform you that your manuscript has been deemed suitable for publication in PLOS ONE. Congratulations! Your manuscript is now being handed over to our production team.

Kind regards, 

on behalf of

Dr. Mehrnaz Kajbafvala 

Academic Editor

PLOS ONE